# Impact of Human Papillomavirus-Negative Dominance in Oropharyngeal Cancer on Overall Survival: A Population-Based Analysis in Germany from 2018 to 2020

**DOI:** 10.3390/cancers15215259

**Published:** 2023-11-02

**Authors:** Mussab Kouka, Laura Gerlach, Jens Büntzel, Holger Kaftan, Daniel Böger, Andreas H. Müller, Thomas Ernst, Orlando Guntinas-Lichius

**Affiliations:** 1Department of Otorhinolaryngology, Jena University Hospital, 07747 Jena, Germany; mussab.kouka@med.uni-jena.de (M.K.); gerlach_laura@web.de (L.G.); 2Department of Otorhinolaryngology, Suedharzklinikum Nordhausen, 99734 Nordhausen, Germany; jens.buentzel@shk-ndh.de; 3Department of Otorhinolaryngology, Helios-Klinikum Erfurt, 99089 Erfurt, Germany; holger.kaftan@helios-gesundheit.de; 4Department of Otorhinolaryngology, SRH Zentralklinikum Suhl, 98527 Suhl, Germany; daniel.boeger@srh.de; 5Department of Otorhinolaryngology, SRH Wald-Klinikum Gera, 07548 Gera, Germany; andreas.mueller@srh.de; 6University Tumor Center, Jena University Hospital, 07747 Jena, Germany; thomas.ernst@med.uni-jena.de

**Keywords:** HPV, oropharyngeal cancer, head and neck cancer, survival

## Abstract

**Simple Summary:**

The impact of the relation of human papillomavirus (HPV) infection and smoking status of oropharyngeal squamous cell carcinoma (OPSCC) on overall survival (OS) was investigated in a retrospective population-based study in Thuringia, Germany. All patients with OPSCC (from 2018 to 2020) were included. OPSCC cases were 37.3% HPV-positive (+) (31.2% smokers; mean incidence: 2.91/100,000 population) and 57.8% HPV-negative (63.5% smokers; mean incidence: 4.50/100,000 population). HPV+ patients had significantly better OS than HPV-negative (−) patients. In multivariable analysis, HPV− patients had an increased 4.5-fold higher hazard of death, but the smoking status had no independent influence on risk of death. In binary logistic regression analysis, smokers showed a 4.5 increased odds ratio (OR) of being tested HPV− than for nonsmokers. HPV− smokers formed the majority in Thuringia. Optimizing OPSCC therapeutic strategies due to the dominance of HPV− is more important than discussing de-escalation strategies for HPV+ patients.

**Abstract:**

The impact of the relation of human papillomavirus (HPV) and smoking status of oropharyngeal squamous cell carcinoma (OPSCC) on overall survival (OS) was investigated in a retrospective population-based study in Thuringia, Germany. A total of 498 patients with OPSCC (76.9% men; mean age 62.5 years) from 2018 to 2020 were included. OPSCC cases were 37.3% HPV-positive (+) (31.2% smokers; mean incidence: 2.91/100,000 population) and 57.8% HPV-negative (63.5% smokers; mean incidence: 4.50/100,000 population). Median follow-up was 20 months. HPV+ patients had significantly better OS than HPV-negative (−) patients (HPV+: 2-year OS: 90.9%; HPV−: 2-year OS: 73.6%; *p* < 0.001). In multivariable analysis, HPV− patients (hazard ratio (HR) = 4.5; 95% confidence interval (CI): 2.4–8.6), patients with higher N classification (N2: HR = 3.3; 95% CI: 1.71–6.20; N3: HR = 3.6; 95% CI: 1.75–7.31) and with a higher cancer staging (III: HR = 5.7; 95% CI: 1.8–17.6; IV: HR = 19.3; 95% CI: 6.3–57.3) had an increased hazard of death. HPV− smokers formed the majority in Thuringia. Nicotine and alcohol habits had no impact on OS. Optimizing OPSCC therapeutic strategies due to the dominance of HPV− is more important than discussing de-escalation strategies for HPV+ patients.

## 1. Introduction

Oropharyngeal squamous cell carcinoma (OPSCC) is increasing in importance among head and neck cancer (HNC), as a significant increase in incidence has been observed, whereas head and neck squamous cell carcinoma (HNSCC) has shown a decreasing trend worldwide. This change in the epidemiology of OPSCC is attributed to a decrease in tobacco use and an increase in exposure to sexually transmitted oral HPV infections [1,2]. With the release of the 8th edition of the TNM classification of malignant tumors in 2017, HPV status is even given its own consideration in OPSCC [3]. Patients with an HPV-associated OPSCC have a better prognosis than patients with classic risk factors and an HPV-negative (−) OPSCC [4]. Compared with HPV− OPSCC patients, HPV-positive (+) OPSCC patients have significantly higher overall survival (OS) and have a mean 30% higher 5-year overall survival (5-OS) [5]. Accordingly, HPV+ OPSCC patients may benefit from de-escalation of treatment. According to recent evidence, bimodal diagnostics (HPV-DNA and p16 expression) are recommended to detect HPV association [6].

According to the Robert Koch Institute (RKI), 9834 men and 4375 women were diagnosed with oral cavity and oropharyngeal carcinomas (C00–C14) in Germany in 2018, based on the International Classification of Diseases for Oncology, 10th revision, German modification (ICD-10-GM) [7]. Male patients are about twice as likely as female patients to develop HNSCC. In addition, female patients have a better 5-year OS rate than male patients. In male patients, incidence rates of HNSCC are declining. In contrast, incidence rates in female patients have remained constant, and in some cases have increased slightly [8]. In Germany, the proportion of HPV+ OPSCC cases is estimated to be between 40% and 50%, which is in line with the general trend in Europe [9,10]. In the UK, the prevalence of HPV+ OPSCC is about 51.8% [11]. In Denmark, HPV+ prevalence is approximately 55%. In a retrospective semi-national registry study in Denmark, a threefold increase in HPV+ OPSCC was observed between 2000 and 2017 [12]. The proportion of HPV+ OPSCC cases in Italy increased from 16.7% in 2000–2006 to 46.1% in 2013–2018 [13]. The highest HPV+ prevalence of OPSCC is reported in the United States, with approximately 70% [14,15]. In addition, the incidence of HPV-related oropharyngeal cancer in the United States differed by age and race. Older white men had the highest incidence rate, increasing to more than 18 per 100,000 in 2015, compared with incidence rates of 6 and 4 per 100,000 for Hispanic and Black men, respectively [16].

In different German studies, the prevalence of HPV+ HNSCC varied from 21% to 53% between 2000 and 2015, and from 38% to 71% between 2004 and 2013 [9,17,18]. The differences in incidence trends result from the retrospective, hospital-based nature of the studies. The calculation of incidence was not based on population-based data, but on data from tumor registries. In addition, prior to the publication of the 8th edition of the TNM classification in 2017, HPV status was not or insufficiently recorded by tumor registries. In previous population-based studies in Germany, HPV+ OPSCC patients showed a 5-year OS of about 80%, whereas the 5-year OS of HPV− OPSCC was only 40–50% [9,18].

To investigate the current impact of HPV association and its incidence, a population-based analysis of cancer registry data of all patients treated for primary OPSCC from 2018 to 2020 was performed. The aim of this retrospective study was to determine the impact of HPV association on patient and tumor characteristics of OPSCC. In addition, the influence of HPV association, considering patient and tumor characteristics, on the OS of OPSCC patients were analyzed.

## 2. Methods

### 2.1. Patients

This population-based, retrospective study was conducted using patient data from the five Thuringian cancer registries (Gera, Erfurt, Nordhausen, Suhl, and Jena). Thuringia is a state in the Federal Republic of Germany and has a population about 2.1 million people. All patients diagnosed with primary OPSCC from January 2018 to December 2020 were included. A total of 498 new cases of primary OPSCC were included. Pathological stages of the primary tumor were recorded using the UICC classification and TNM classification (7th edition) [19]. The standard data set of the cancer registries was used. This data set included patients’ characteristics, tumor characteristics including HPV data, the tumor classification as well as uniform data on the treatment. The HPV test material was derived from the primary tumors. Diagnostics were performed by polymerase chain reaction (PCR) and chip hybridization or by immunohistochemical detection of p16 [20]. To cross-check the validity of the HPV data and to collect additional data on smoking habits and alcohol consumption (both not assessed on the cancer registries), the individual data sets of the five cancer registries were deblinded within each of the centers. Then, a crosscheck with each patient chart could be performed, i.e., a data linkage between the cancer registry data (population-based data) and the patient chart in the treating hospital (hospital-based data) could be performed. After this process, the complete data set was re-anonymized and evaluated.

### 2.2. Statistical Analysis

SPSS Statistics version 29.0 (IBM Deutschland GmbH, 71139 Ehningen, Germany) was used to perform the statistical analyses. Epidemiological measures were calculated from absolute case numbers and data from the Thuringian State Office of Statistics (https://www.statistik.thueringen.de/; accesses on 30 March 2023) on the population of Thuringia during the observation period. Frequency analyses and cross-tabulations were performed. Furthermore, the chi-square test was performed to analyze ordinal and nominal data. Binary logistic regression analyses were performed to examine the association of HPV status with patients’ characteristics, tumor characteristics and treatment characteristics. Results for independent factors are reported as odds ratios (OR) with 95% confidence intervals (CI) and a significance level of *p* < 0.05. For analysis of survival time data, significant factors from univariate analysis were included in multivariate analyses using Cox proportional hazard ratio (HR) with corresponding 95% CI. Kaplan–Meier calculations were performed to estimate the impact of the variables on OS. The significance level was set to *p* < 0.05.

### 2.3. Ethical Considerations

The Ethics Committee of the Jena University Hospital approved the retrospective study (IRB No. 2018-1075-Material; 2022-2526-BO). Informed consent of the patients was waived, as this study had a non-interventional retrospective design, and all data were analyzed anonymously.

## 3. Results

### 3.1. Patients’ Characteristics and Tumor Characteristics

The distribution of patient and tumor characteristics of oropharyngeal carcinomas in Thuringia in the period 2018–2020 is shown in Table 1. A total of 498 patients were included in the study. Of those, 383 (76.9%) were male and 115 (23.1%) were female patients. The mean age was 62.5 ± 9.4 years at the time of diagnosis (median: 61 years). The most common sub-localization of OPSCC was in the tonsil (43.8%). Most patients (288 patients; 57.8%) were HPV−. About one third of patients (186 patients; 37.3%) were HPV+. The proportion of smokers was 51.4% (256). Further relationships between HPV status and smoking habits are shown in Figure 1. The proportion of patients who regularly consumed alcohol was 34.9%. Most patients were in the highest UICC tumor stage, stage IV (40.4%). According to the T classification, most patients were classified as stage T4 (145 patients; 29.3%). At the time of diagnosis, no lymph node involvement (N0) was detected in 131 patients (26.5%). An N1 stage was assigned to 91 patients (18.4%). Most patients were classified as stage N2 (205 patients, 41.4%). Most patients (455 patients; 91.9%) had no distant metastases (M0) at the time of diagnosis. Many patients (254 patients; 51%) were graded as type G2 (moderately differentiated) for tumor grading, and less than one-third (143 patients; 28.7%) were graded as G3 (poorly differentiated). Radiotherapy was the most common treatment modality used (385 patients, 77.3%). Surgical therapy was used in half of the cases (276 patients, 55.4%). Immunotherapy was used in only 89 patients (17.9%).

### 3.2. Associations of HPV Status with Patient and Tumor Characteristics

The correlations between an HPV association and patient and tumor characteristics are shown in Table 2. There was no significant association with age, alcohol drinking, recurrence, radiotherapy, and chemotherapy (all *p* > 0.5). In contrast, gender, smoking status, localization, T classification, N classification, M classification, UICC stage, grading, surgery, and immunotherapy showed a significant correlation with HPV status (all *p* < 0.5). Thus, male patients (80.2%) and patients with smoking habits (63.5%) were predominantly HPV−. Patients with the sub-localization oropharynx without any specification (27.8%) and soft palate (12.5%) were predominantly HPV−. Regarding tumor characteristics, patients with T3 (25.35%) and T4 (34.72%) classification, N2 (41.67%) and N3 (15.97%) classification and M1 (11.81%) classification were also predominantly HPV−. Patients treated surgically (50.0%) or not receiving immunotherapy (77.1%) were also predominantly HPV−. Patients with moderately differentiated carcinomas (G2 grading) were predominantly HPV− (56.94%).

### 3.3. Binary Logistic Regression of HPV Status to Patient and Tumor Characteristics

The odds ratio (OR) and 95% CI for HPV-negative association are shown in Table 3. Model 1 grouped patient, staging and treatment characteristics, whereas in model 2, treatment characteristics were excluded. Smokers showed a 4.5-fold increased OR of being tested HPV− than nonsmokers (OR: 4.5; 95% CI: 2.76–7.35; *p* < 0.001). The sub-localizations soft palate and oropharynx without any specification were associated with a higher risk of being HPV− than tonsils (soft palate: OR: 5.5; 95% CI: 1.71–17.77, *p* = 0.004; oropharynx without any specification: OR: 2.2; 95% CI: 1.20–4.20, *p* = 0.012). Patients with metastases were 6.1 times more likely to be tested HPV-negative than patients without metastases (OR: 6.1; 95% CI: 1.30–28.97; *p* = 0.022). With additional consideration for cancer staging and treatment characteristics in model 3, cancer staging III and IV were also associated with a higher risk of being HPV− (stage III: OR: 5.4; 95% CI: 2.60–11.31, *p* < 0.001; stage IV: OR: 274.3; 95% CI: 85.77–877.34, *p* < 0.001). Patients who underwent surgical treatment were 2.4 times more likely to test negative for HPV (OR: 2.4; 95% CI: 1.23–4.57; *p* = 0.010). Gender, T classification, N classification, grading and immunotherapy showed no significance.

### 3.4. Influence of HPV Status and Other Factors on Overall Survival

The median follow-up of all patients was 20 months. A total of 106 patients (21.3%) died during the observation period. For the 392 patients alive, the median follow-up was 24 months (Appendix A). OS was worse in patients without HPV association (HPV−: 2-year OS: 73.6%, mean survival of 36.3 months; 95% confidence interval (CI): 33.65–38.94) than in patients with HPV association (HPV+: 2-year OS: 90.9%, mean survival of 47.18 months; 95% CI: 45.07–48.29; *p* < 0.001; Appendix A). Kaplan–Meier calculations according to gender, cancer staging, HPV status, smoking status and alcohol drinking habits are shown in Figure 2. The 2-year survival rate was 77.8% for male patients and 86.1% for female patients (*p* < 0.040). Smoking and alcohol abuse were correlated with significantly lower OS (2-year OS = 75.39% smokers and 85.1% non-smokers; 2-year OS = 75.3% alcoholics and 83.5% non-alcoholics). The extent of T classification had a significant effect on OS (*p* < 0.001). The 2-year survival rate was 88.2% for T1 and 86.7% for T2. In contrast, the 2-year survival rates for T3 and T4 were 78.5% and 70.3%, respectively. Lymph node involvement also had a significant impact on overall survival (2-year OS: N0 = 87.8%; N1 = 86.8%; N2 = 75.6%; N3 = 67.7%; *p* < 0.001). The development of distant metastases (M) significantly affected survival (*p* < 0.001). There was significantly lower OS in patients with advanced cancer staging (*p* = 0.001). UICC stage I patients had a 2-year survival rate of 96.3%, and stage II patients, 90.9%. In contrast, locally advanced stage III and IV patients had a rate of 83.9% and 67.5%, respectively.

### 3.5. Multivariable Analysis of Factors Influencing Overall Survival

Table 4 shows the multivariable analysis for patient, staging and treatment characteristics. Several patient and staging characteristics were grouped in model 1, and staging and treatment characteristics were grouped in model 2. HPV− patients had a significantly higher mortality rate (hazard ratio (HR) = 4.5; 95% CI: 2.36–8.59; *p* < 0.001). Gender, age, nicotine and alcohol habits and T and M classification had no significant effect on overall survival (all *p* > 0.005). In model 1 and model 2, higher N classification correlated with a significantly higher mortality rate (model 2: N2: HR = 3.3; 95% CI: 1.71–6.20; *p* < 0.001; N3: HR = 3.6; 95% CI: 1.75–7.31; *p* < 0.001). With additional consideration for cancer staging in model 3, patients with cancer staging III and IV had significantly worse survival (III: HR = 5.7; 95% CI: 1.82–17.56; *p* = 0.003; IV: HR = 19.3; 95% CI: 6.52–57.32; *p* < 0.001). Patients who did not receive any kind of treatment (primary surgery, radiation, or chemotherapy) had an increased risk of death. Patients without primary surgery had 2.5 times greater risk of death than patients with primary surgery (HR = 2.5; 95% CI: 1.54–3.92; *p* < 0.001) and without radiation had a 5.6-fold higher risk of death than patients treated with radiation (HR = 5.6; 95% CI: 3.18–9.74; *p* < 0.001). Patients without chemotherapy had a 1.7-fold greater risk of dying (HR = 1.7; 95% CI: 1.05–2.97; *p* = 0.033) than patients with chemotherapy.

### 3.6. Incidence of OPSCC

On 31 December 2018, 2,143,145 people lived in Thuringia. On 31 December 2019 and 2020, 2,133,378 and 2,120,237 people lived in Thuringia, respectively. The crude incidence rate for patients with OPSCC in Thuringia was 7.56 in 2018, 8.25 in 2019 and 7.55 in 2020 per 100,000 population (Table 5 and Figure 3). The incidence of OPSCC in male patients (2018: 11.88; 2019: 12.78; 2020: 11.63 per 100,000 population) was higher than that in female patients (2018: 3.33; 2019: 3.81; 2020: 3.55 per 100,000 population). Accordingly, HPV+ and HPV− incidence was also higher in men than in women. The incidence among smokers was higher in all years (2018: 3.92; 2019: 4.27; 2020: 3.82 per 100,000 population) than among nonsmokers (2018: 3.08; 2019: 3.75; 2020: 3.25 per 100,000 population). The incidence among smokers with HPV+ was 0.56 in 2018, 1.03 in 2019 and 1.13 in 2020 per 100,000 population. In comparison, the incidence among non-smoking patients with HPV+ was higher than that among smokers with HPV+ (1.87 in 2018, 1.83 in 2019 and 1.84 in 2020 per 100,000 population). HPV− smokers had the highest incidence rate (2.99 in 2018, 3.14 in 2019 and 2.45 in 2020 per 100,000 population). Regarding alcohol drinking, the incidence rate among patients that consumed alcohol was lower in all years (2018: 2.47; 2019: 2.53; 2020: 3.16 per 100,000 population) than that in non-alcoholics (2018: 4.34; 2019: 5.39; 2020: 3.87 per 100,000 population).

## 4. Discussion

HPV association indicates a better prognosis in OPSCC compared to patients with the classical risk factors and an HPV− tumor. However, the results of our population-based study showed that HPV− status was dominant and had the greatest impact on OS here. In contrast to the Anglo-Saxon countries, the proportion of classical risk factors, especially smoking, remains very high in Germany for HNC [21]. Therefore, consideration of the association of HPV status and smoking status in a population-based analysis is very important. The results of binary logistic regression analysis showed that smoking, M1 classification, and higher cancer staging were associated with a higher chance of being HPV−. HPV− patients were more frequently treated with primary surgery. There was no significant association with age, alcohol drinking, recurrence, or decision to use radiation and chemotherapy. In multivariate analysis, negative, significantly associated factors for OS were HPV−, higher N classification (N2/3), M1 classification, higher cancer staging (UICC III/IV), and no treatment (primary surgery, radiation, and chemotherapy). Wagner et al. reported a similar finding in their population-based study in another federal state in Germany [22]. Wagner et al. showed that HPV status, tobacco smoking, T and N classification and age were significant predictors of OS in univariate analysis, and except for smoking in multivariate analysis. According to Wagner et al., HPV status was more important for OS in OPSSC than smoking. These results are consistent with our findings. Gillison et al. indicated that there is an association between HPV− and nicotine and alcohol abuse [5,15]. These findings are consistent with Yin et al., who reported an association between lower OS in HPV− OPSCC and smoking, but the association between smoking and lower OS in HPV− OPSCC was not statistically significant (*p* = 0.14) [23]. Alcohol consumption, on the other hand, showed no significant correlation with HPV status in our study. In the literature, however, alcohol consumption and HPV+ status are synergistically associated with OPSCC [24,25].

Accordingly, the risk factor HPV had a higher influence on the mortality risk of the Thuringian patients than the classical risk factor nicotine consumption. Nevertheless, the proportion of smoking patients in this population should be noted, as it was slightly more than half (51.4%). In other countries, the prevalence of smoking has been significantly reduced through tobacco control programs [26,27]. In the literature, there is a considerable discussion regarding de-escalation strategies for HPV+ OPSCC [28,29,30]. However, our results indicate that HPV− status has the greatest impact on patient OS, and treatment optimization of HPV− OPSCC should be considered in further studies.

The proportion of tonsil cancer to OPSCC was greatest in this study and was associated with HPV+ status. A Swedish study by Nasman et al. reported a rising incidence of tonsillar squamous cell carcinoma that was also associated with HPV+ status [31].

As in many other studies, HPV+ patients showed significantly better OS than HPV− patients in the present study. A comparison of mean OS time highlights the better prognosis in HPV+ patients.

The incidence of HPV+ OPSCC in this population-based study was 2.61 per 100,000 population with a proportion of 37.3% HPV+ OPSCC. In contrast, a previous study from the same population showed a much lower proportion of HPV+ OPSCC (17.2%) with an incidence of 1.89 per 100,000 population. Population-based data are sparse. The population-based studies by Wagner et al. and Wittekindt et al. reported a lower proportion of HPV+ OPSCC in another federal state in Germany (20.6% and 27.1%, respectively) [9,22]. In contrast, European comparisons showed significantly higher proportions of HPV+ OPSCC cases. In the UK, the prevalence of HPV+ OPSCC cases between 2002 and 2011 was 51.8%, in Denmark, HPV+ OPSCC accounted for approximately 55% of all cases between 2000 and 2017, and in Italy HPV+ OPSCC cases increased from 16.7% in 2000–2006 to 46.1% in 2013–2018 [11,12,13]. In their retrospective analysis of 730 OPSCC patients, Wittekindt et al. recorded an increase in the incidence of OPSCC for the period 1999–2014, from 6.2 to about 6.8 per 100,000 population [9]. The Danish study showed an age-adjusted increase in the incidence of OPSCC, from 1.8 in 2000 to 5.1 in 2017 per 100,000 population [12]. This is consistent with the age-adjusted incidence of OPSCC across Germany (2018: 5.1 and 2019: 5.0 per 100,000 population).

In a study of the same federal state during a treatment period from 1996 to 2016, Dittberner et al. reported an increase in the incidence of OPSCC patients with HPV-independent HNSCC (crude incidence rate: 15.98 per 100,000 population). HPV status was not examined, but the incidence increase was attributed to the possibility of increased HPV prevalence in OPSCC [32].

The present study was limited by its retrospective character and the only 24 months of follow-up. Pre-existing conditions and comorbidities could not be considered in more detail, as they are only partially recorded in the cancer registry. In addition, the coding for the sub-localization oropharynx without any specification might be inaccurate. Whether the tumor has spread over one region or over several localizations was unknown. Furthermore, the exact amount of nicotine and alcohol consumption was unknown. Data on existing nicotine and alcohol consumption were not sufficient. An exact indication of the amount of nicotine consumption in pack years and alcohol consumption could be more relevant. One advantage of the study is the uniform detection method for HPV association since the introduction of the TNM classification in the 8th edition of 2017. Since the vast majority of patients with OPSCC were HPV−, better treatment strategies are needed for this population. This is more important than a discussion of de-escalation strategies for HPV+ patients. And although nicotine use did not show a significant impact on OS in multivariable analyses, anti-smoking education campaigns continue to be very important. More epidemiologic studies of HPV-associated OPSCC with larger populations over a longer study period are needed to show the incidence trend.

## 5. Conclusions

This retrospective, population-based study from 2018 to 2020 in Thuringia describes the dominance of HPV− status in OPSCC. According to this study, HPV+ OPSCC patients were a minority in Thuringia. Smokers, patients with an OPSCC without any specification and soft palate, M1 classification and UICC stage III/IV had a higher chance of being HPV−. In multivariable analysis, HPV− status, N2/3 classification, and higher cancer staging were associated with lower OS. Classical risk factors continue to dominate and have the greatest impact on OS. Therefore, better therapeutic strategies for HPV− OPSCC patients should be developed for this population. Optimizing therapeutic strategies due to the dominance of HPV− status is more important than discussing de-escalation strategies for HPV+ OPSCC patients.

## Figures and Tables

**Figure 1 cancers-15-05259-f001:**
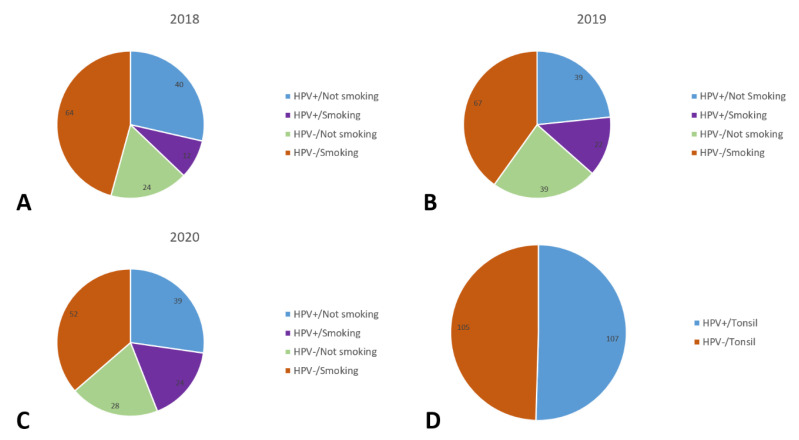
Distribution of patients by HPV-positive (HPV+) and HPV-negative (HPV−) and smoking habits for 2018 (**A**), 2019 (**B**), and 2020 (**C**), and extra for the tonsil sub-localization (**D**).

**Figure 2 cancers-15-05259-f002:**
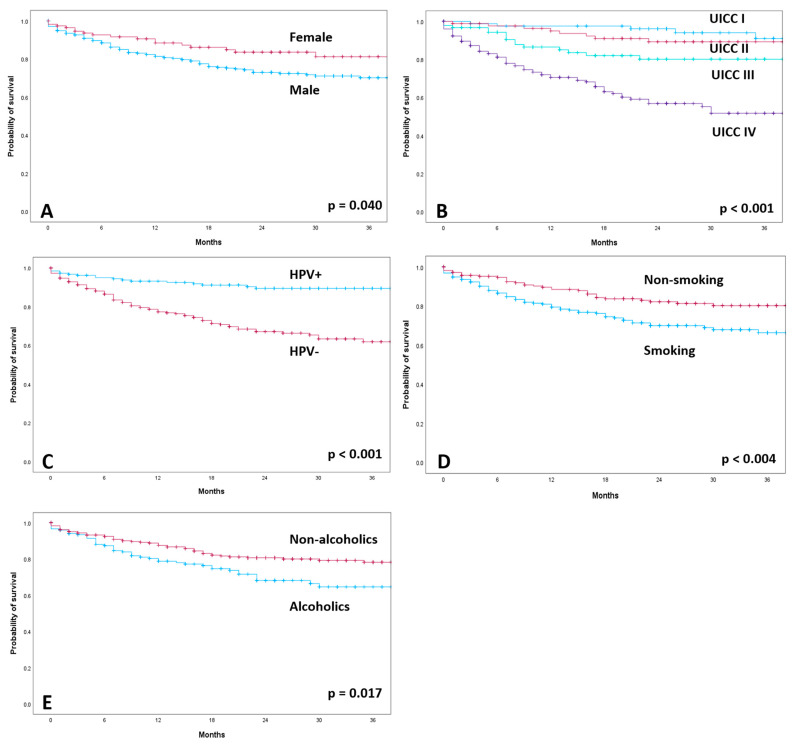
Kaplan–Meier curves of overall survival according to gender (**A**), cancer staging (**B**), HPV-positive (HPV+) and HPV-negative (HPV−) (**C**), smoking status (**D**) and alcohol drinking habits (**E**).

**Figure 3 cancers-15-05259-f003:**
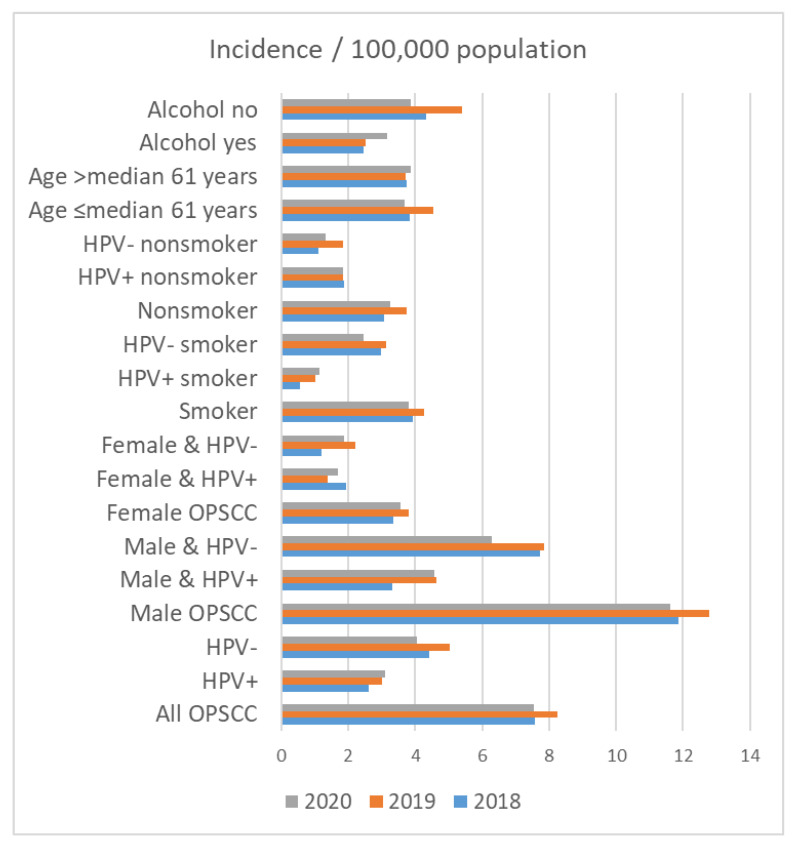
Incidence of oropharyngeal cancer (OPSCC) in Thuringia per 100,000 population in the years 2018, 2019, and 2020 in relation to patient characteristics.

**Table 1 cancers-15-05259-t001:** Patients’ characteristics, histopathology, and treatment characteristics.

Parameter	Frequency (N)	%
Gender		
Male	383	76.9
Female	115	23.1
Localization		
Oropharynx without specification	122	24.5
Tonsil	218	43.8
Base of tongue	116	23.3
Soft palate	42	8.4
HPV status (HPV-DNA or p16) *		
HPV-positive	186	37.3
HPV-negative	288	57.8
Unknown	24	4.8
Cigarette smoking		
Yes	256	51.4
No	215	43.2
Unknown	27	5.4
Alcohol drinking		
Yes	174	34.9
No	290	58.2
Unknown	34	6.8
Cancer staging		
0	1	0.2
I	82	16.6
II	88	17.8
III	93	18.8
IV	200	40.4
Unknown	31	6.3
T classification		
Carcinoma in situ	1	0.2
T1	92	18.6
T2	128	25.9
T3	121	24.4
T4	145	29.3
TX	8	1.6
N classification		
N0	131	26.5
N1	91	18.4
N2	205	41.4
N3	62	12.5
NX	6	1.2
M classification		
M0	455	91.9
M1	40	8.1
Recurrence		
Yes	69	13.9
No	278	55.8
Unknown	151	30.3
Grading		
G1	17	3.4
G2	254	51.0
G3	143	28.7
Undifferentiated	36	7.2
Unknown	48	9.6
Treatment		
Best supportive care (BSC)	31	6.2
Surgery **	276	55.4
Radiation	385	77.3
Chemotherapy	289	58.0
Immunotherapy	89	17.9
Unknown	5	1.0
	Mean ± SD	Median, Range
Age in years	62.45 ± 9.5	61, 33–95

** ± adjuvant therapy, HPV-DNA = human papillomavirus-deoxyribonucleic acid, which was detected by polymerase chain reaction (PCR), p16 = immunohistochemical evidence of p16INK4a expression, * in case of discrepancy, HPV-DNA was preferred over p16 status.

**Table 2 cancers-15-05259-t002:** Correlations between HPV status and patients’ characteristics, histopathology and treatment.

Parameter		HPV-Positive, N = 186		HPV-Negative, N = 288		Total, N = 498		*p*
		N	(%)	N	(%)	N	(%)	
Gender	Male	132	71.0	231	80.2	363	72.9	**0.020**
	Female	54	29.0	57	19.8	111	22.3	
Age, median	≤61 years	90	48.4	153	53.1	243	48.8	0.314
	>61 years	96	51.6	135	46.9	231	46.4	
Cigarette smoking	Yes	58	31.2	183	63.5	241	48.4	**<0.001**
	No	118	63.4	91	31.6	209	42.0	
Alcohol drinking	Yes	57	30.6	108	37.5	165	33.1	0.106
	No	118	63.4	161	55.9	279	56.0	
Localization	Oropharynx without specification	33	17.7	80	27.8	113	22.7	**<0.001**
	Tonsil	107	57.5	105	36.5	212	42.6	
	Base of tongue	40	21.5	67	23.3	107	21.5	
	Soft palate	6	3.2	36	12.5	42	8.4	
T classification	T1	46	24.73	46	15.97	92	18.47	**<0.001**
	T2	62	33.33	64	22.22	126	25.30	
	T3	39	20.97	73	25.35	112	22.49	
	T4	35	18.82	100	34.72	135	27.11	
N classification	N0	46	24.73	80	27.78	126	25.30	**<0.001**
	N1	51	27.42	36	12.50	87	17.47	
	N2	75	40.32	120	41.67	195	39.16	
	N3	13	6.99	46	15.97	59	11.85	
M classification	M0	180	96.77	254	88.19	434	87.15	**<0.001**
	M1	5	2.69	34	11.81	39	7.83	
Cancer staging	Stage I	64	34.41	18	6.25	82	16.47	**<0.001**
	Stage II	65	34.95	23	7.99	88	17.67	
	Stage III	47	25.27	46	15.97	93	18.67	
	Stage IV	5	2.69	195	67.71	200	40.16	
Grading G	G1	4	2.15	13	4.51	17	3.41	**0.007**
	G2	75	40.32	164	56.94	239	47.99	
	G3	52	27.96	87	30.21	139	27.91	
	Undifferentiated	20	10.75	13	4.51	33	6.63	
Primary surgery	Yes	128	68.8	144	50.0	272	54.6	**0.001**
	No	58	31.2	139	48.3	197	39.6	
Radiation	Yes	151	81.2	219	76.0	370	74.3	0.324
	No	35	18.8	64	22.2	99	19.9	
Chemotherapy	Yes	110	59.1	164	56.9	274	55.0	0.798
	No	76	40.9	119	41.3	195	39.2	
Immunotherapy	Yes	23	12.4	61	21.2	84	16.9	**0.011**
	No	163	87.6	222	77.1	385	77.3	
Recurrence	Yes	21	11.3	46	16.0	67	13.5	0.153
	No	165	88.7	242	84.0	407	81.7	

HPV = human papillomavirus; significant *p*-values (*p* < 0.05) in bold.

**Table 3 cancers-15-05259-t003:** Binary logistic regression models of associations with HPV-negative status.

Parameter		OR	95% CI	*p*
Model 1: Patient, staging and treatment characteristics				
Gender	Female	1	Reference	0.906
	Male	1.0	0.59–1.83	
Cigarette smoking	No	1	Reference	**<0.001**
	Yes	4.5	2.76–7.35	
Localization	Tonsil	1	Reference	
	Base of tongue	1.5	0.80–2.70	0.216
	Oropharynx without specification	2.2	1.20–4.20	**0.012**
	Soft palate	5.5	1.71–17.77	**0.004**
T classification	T1	1	Reference	
	T2	0.7	0.39–1.54	0.462
	T3	0.9	0.43–1.98	0.839
	T4	1.0	0.45–2.42	0.931
N classification	N0	1	Reference	
	N1	0.7	0.34–1.37	0.282
	N2	1.0	0.53–1.76	0.895
	N3	1.3	0.51–3.08	0.617
M classification	M0	1	Reference	**0.022**
	M1	6.1	1.30–28.97	
Grading	G1	1	Reference	
	G2	0.5	0.13–2.12	0.361
	G3	0.5	0.12–2.08	0.341
	Undifferentiated	0.3	0.06–1.39	0.119
Primary surgery	Yes	1	Reference	
	No	1.6	0.9–2.9	0.145
Immunotherapy	No	1	Reference	
	Yes	1.2	0.64–2.38	0.534
Model 2: Patient and staging characteristics				
Gender	Female	1	Reference	0.919
	Male	1.0	0.59–1.81	
Cigarette smoking	No	1	Reference	**<0.001**
	Yes	4.2	2.61–6.87	
Localization	Tonsil	1	Reference	
	Base of tongue	1.5	0.83–2.76	0.178
	Oropharynx without specification	2.3	1.2–4.20	**0.010**
	Soft palate	5.5	1.71–17.69	**0.004**
T classification	T1	1	Reference	
	T2	0.8	0.41–1.63	0.570
	T3	1.1	0.52–2.24	0.840
	T4	1.4	0.68–3.05	0.335
N classification	N0	1	Reference	
	N1	0.6	0.31–1.26	0.188
	N2	1.0	0.54–1.76	0.938
	N3	1.3	0.56–3.25	0.513
M classification	M0	1	Reference	**0.015**
	M1	6.8	1.45–31.73	
Grading	G1	1	Reference	
	G2	0.6	0.14–2.14	0.386
	G3	0.5	0.13–2.04	0.342
	Undifferentiated	0.3	0.06–1.37	0.116
Model 3: Staging and treatment characteristics				
Cancer staging	I	1	Reference	
	II	1.5	0.72–3.12	0.280
	III	5.4	2.60–11.31	**<0.001**
	IV	274.3	85.77–877.34	**<0.001**
Primary surgery	No	1	Reference	**0.010**
	Yes	2.4	1.23–4.57	
Immunotherapy	Yes	1	Reference	0.466
	No	1.4	0.60–3.02	

OR—Odds ratio; CI—Confidence interval; significant *p*-values (*p* < 0.05) in bold.

**Table 4 cancers-15-05259-t004:** Multivariable Cox regression analysis for associations with lower overall survival.

Parameter		HR	95% CI	*p*
Model 1: Patient and staging characteristics				
Gender	Female	1	Reference	0.803
	Male	1.1	0.58–2.02	
Age	≤61 years	1	Reference	0.064
	>61 years	1.6	0.98–2.47	
Cigarette smoking	No	1	Reference	0.748
	Yes	1.1	0.65–1.83	
Alcohol drinking	No	1	Reference	0.168
	Yes	1.4	0.87–2.27	
HPV status	HPV+	1	Reference	**<0.001**
	HPV−	4.5	2.36–8.59	
T classification	T1	1	Reference	
	T2	1.0	0.46–2.07	0.953
	T3	1.1	0.52–2.32	0.817
	T4	1.9	0.93–3.79	0.078
N classification	N0	1	Reference	
	N1	1.3	0.61–2.95	0.467
	N2	2.0	1.11–3.76	**0.022**
	N3	3.2	1.57–6.57	**0.001**
M classification	M0	1	Reference	0.909
	M1	1.0	0.52–2.10	
Model 2: Staging and treatment characteristics I				
T classification	T1	1	Reference	
	T2	1.1	0.50–2.21	0.898
	T3	1.6	0.77–3.50	0.205
	T4	2.1	0.96–4.74	0.063
N classification	N0	1	Reference	
	N1	1.7	0.80–3.71	0.168
	N2	3.3	1.71–6.20	**<0.001**
	N3	3.6	1.75–7.31	**<0.001**
M classification	M0	1	Reference	0.093
	M1	1.7	0.92–3.05	
Primary surgery	Yes	1	Reference	**0.007**
	No	2.2	1.24–3.81	
Radiation	Yes	1	Reference	**<0.001**
	No	4.0	2.31–6.75	
Chemotherapy	Yes	1	Reference	**0.009**
	No	1.9	1.18–3.14	
Model 3: Staging and treatment characteristics II				
Cancer staging	I	1	Reference	
	II	2.7	0.81–9.10	0.104
	III	5.7	1.82–17.56	**0.003**
	IV	19.3	6.52–57.32	**<0.001**
Primary surgery	Yes	1	Reference	
	No	2.5	1.54–3.92	**<0.001**
Radiation	Yes	1	Reference	
	No	5.6	3.18–9.74	**<0.001**
Chemotherapy	Yes	1	Reference	**0.033**
	No	1.7	1.05–2.97	

Significant *p*-values (*p* < 0.05) in bold.

**Table 5 cancers-15-05259-t005:** Incidence of oropharyngeal carcinoma (OPSCC) in Thuringia from 2018 to 2020.

		2018		2019		2020	
		*N*	Incidence/ 100.000	*N*	Incidence/ 100.000	*N*	Incidence/ 100.000
OPSCC		162	7.56	176	8.25	160	7.55
HPV+		56	2.61	64	3.00	66	3.11
HPV−		95	4.43	107	5.02	86	4.06
Male		126	11.88	135	12.78	122	11.63
HPV+		35	3.30	49	4.64	48	4.57
HPV−		82	7.73	83	7.86	66	6.29
Female		36	3.33	41	3.81	38	3.55
HPV+		21	1.94	15	1.39	18	1.68
HPV−		13	1.20	24	2.22	20	1.87
Cigarette smoking		84	3.92	91	4.27	81	3.82
HPV+		12	0.56	22	1.03	24	1.13
HPV−		64	2.99	67	3.14	52	2.45
Non-smoking		66	3.08	80	3.75	69	3.25
HPV+		40	1.87	39	1.83	39	1.84
HPV−		24	1.12	39	1.83	28	1.32
Age	≤Median 61 years	82	3.83	97	4.55	78	3.68
	>Median 61 years	80	3.73	79	3.70	82	3.87
Alcohol drinking	Yes	53	2.47	54	2.53	67	3.16
	No	93	4.34	115	5.39	82	3.87

## Data Availability

The datasets used during the current study are available from the corresponding author upon reasonable request.

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
