# Peer review of "Impact of Human Papillomavirus-Negative Dominance in Oropharyngeal Cancer on Overall Survival: A Population-Based Analysis in Germany from 2018 to 2020"

_cancers, 2023, doi:10.3390/cancers15215259_

Round 1

Reviewer 1 Report

Comments and Suggestions for Authors

The article outlines the impact of the relation of HPV infection and smoking status of oropharyngeal squamous cell carcinoma on overall survival in a population-based study in Thuringia, Germany. A good number of patient cohort was selected and studied over the period of 2018 to 2020. Data has been collected and presented in a comprehensive manner and the article is well-structured. The article can be accepted with minor revisions.

Minor corrections:

Line 84: OPSCC in 2018 instead of 20018

Line 161: t

Line 190: were 

Comments on the Quality of English Language

The article can be accepted with minor revisions.

Author Response

Reviewer's comments:

The article outlines the impact of the relation of HPV infection and smoking status of oropharyngeal squamous cell carcinoma on overall survival in a population-based study in Thuringia, Germany. A good number of patient cohort was selected and studied over the period of 2018 to 2020. Data has been collected and presented in a comprehensive manner and the article is well-structured. The article can be accepted with minor revisions.

1.1 Minor corrections:

Line 84: OPSCC in 2018 instead of 20018

Line 161: t

Line 190: were 

Answer 1.1: We have corrected the typos.

Reviewer 2 Report

Comments and Suggestions for Authors

This study, while interesting, suffers from a lack of rigor in description of these methods. This part needs to be improved. 

In addition, the results section is too dense and a little confusing, and needs to be reworked too.

My comments can be found directly in the PDF

Reviewer 3 Report

Comments and Suggestions for Authors

In this interesting retrospective study, the impact of the incidence of HPV infection and smoking on oropharyngeal squamous cell carcinoma (OPSCC) overall survival was investigated in a cohort of OPSCC patients Thuringia, Germany.  In a well-controlled and multivariable study, a population-based analysis of cancer registry data of all patients treated for OPSCC between 2018 and 2020 was performed.  OPSCC patients were 37.3% HPV-positive (31.2% smokers) and 57.8% HPV-negative 63.5% smokers).   In quite unexpected findings, HPV+ patients had a significantly better overall survival rate than HPV- patients.  Also, HPV- patients had a more than four-fold higher death rate than HPV+ patients, but neither smoking nor alcohol had any significant effect on the risk of death.  Based on these findings, the authors suggest that OPSCC therapeutic studies should be directed more at HPV- patients than de-escalation strategies for HPV+ patients.

This is considered an important, perhaps game-changing study that should be taken into account in the treatment of OPSCC patients.  At the very least, more attention should be devoted to HPV- patients, as their prognosis is definitely worse.  The manuscript is extremely well written with no weaknesses or discrepancies and is suitable for publication.

Comments on the Quality of English Language

English usage requires significant attention.

Author Response

Thank for your supporting feedback

Round 2

Reviewer 2 Report

Comments and Suggestions for Authors

Thanks to the authors for their additions and explanation. this article seems acceptable to me in this form.